# Onset of strong Iceland-Scotland overflow water 3.6 million years ago

Matthias Sinnesael [1,2,37] ✉, Boris-Theofanis Karatsolis[3,4,37], Paul N. Pearson [5], Anne Briais[6], Sidney R. Hemming [7,8], Leah J. LeVay [9], Tom Dunkley Jones [10], Ying Cui [11], Anita Di Chiara [12], Justin P. Dodd[13], Deepa Dwyer[14], Deborah E. Eason [15], Sarah A. Friedman [16], Emma Hanson [10], Katharina Hochmuth [17,18], Halima E. Ibrahim [19], Claire E. Jasper [7,8], Saran Lee-Takeda [20], Danielle E. LeBlanc [21], Melody R. Lindsay [22], David D. McNamara [23], Sevasti E. Modestou [24], Margaret A. Morris [25], Bramley J. Murton[26], Suzanne OConnell [27], Gabriel Pasquet [28], Sheng-Ping Qian[29], Yair Rosenthal [30], Sara Satolli [31], Takuma Suzuki[32], Thena Thulasi [33], Bridget S. Wade [5], Nicholas J. White[34], Tao Wu[35], Alexandra Y. Yang [36] & Ross E. Parnell-Turner [25]

North Atlantic Deep Water (NADW), the return flow component of the Atlantic Meridional Overturning Circulation (AMOC), is a major inter-hemispheric ocean water mass with strong climate effects but the evolution of its source components on million-year timescales is poorly known. Today, two major NADW components that flow southward over volcanic ridges to the east and west of Iceland are associated with distinct contourite drift systems that are forming off the coast of Greenland and on the eastern flank of the Reykjanes (mid-Atlantic) Ridge. Here we provide direct records of the early history of this drift sedimentation based on cores collected during International Ocean Discovery Programme (IODP) Expeditions 395C and 395. We find rapid acceleration of drift deposition linked to the eastern component of NADW, known as Iceland–Scotland Overflow Water at 3.6 million years ago (Ma). In contrast, the Denmark Strait Overflow Water feeding the western Eirik Drift has been persistent since the Late Miocene. These observations constrain the long-term evolution of the two NADW components, revealing their contrasting independent histories and allowing their links with climatic events such as Northern Hemisphere cooling at 3.6 Ma, to be assessed.

The North Atlantic Current (NAC) carries warm high-salinity waters from the Gulf Stream to the European continental margin. Some of this water mass penetrates as far north as the Norwegian Sea, where it loses residual heat, densifies and facilitates down-welling to form a cold, deep return flow across the Iceland-Scotland Ridge known as Iceland–Scotland Overflow Water (ISOW) (Fig. 1a). Bottom-water contour currents, so-called because they roughly flow along seafloor contours based on their density, are directed by bathymetry and deflected by the Coriolis effect. Consequently, ISOW initially flows south along the eastern flank of the Reykjanes Ridge, then crosses the ridge in deep fracture zones to turn north into the western North Atlantic. Here it meets the Denmark Strait Overflow Water (DSOW), a cold deep-water current that flows south from the Arctic and through the Denmark Strait between Greenland and Iceland (Fig. 1a). The two

---

**Fig. 1 | Ocean circulation, bathymetry, and boreholes in the North Atlantic Ocean and Nordic Seas.** Coloured shaded circles = Expeditions 395C and 395 sites; black dots = Ocean Drilling Program sites. **a** Map: Grey shading = areas < 500 m below sea level; pink/blue arrows = near-surface / deep-water overflow paths, respectively (ref. 57); brown polygons = selected contourite drifts (ref. 12); DS = Denmark Strait; DSOW= Denmark Strait Overflow Water; ISOW = Iceland–Scotland Overflow Water (dark blue arrows); ISR = Iceland–Scotland Ridge; NAC = North Atlantic Current; NADW = North Atlantic Deep Water. **b** Plate spreading flowline transect intersecting 395/395C sites, based on seismic reflection profile (ref. 41); grey shading = sediment.

currents combine and flow into the Labrador Sea to produce North Atlantic Deep Water (NADW). Taken together, the system of warm saline surface waters that flow northward, cool and sink, before returning southward as NADW is known as the Atlantic Meridional Overturning Circulation (AMOC).

The dynamics of the AMOC are highly sensitive to changes in temperature, salinity, sea ice cover, and seafloor topography[1,2]. Past changes in AMOC have been linked to both long- and short-term climatic oscillations, with NADW transport thought to have been stronger during interglacial periods as well as in the warm Early Pliocene[3,4], although several lines of evidence predict a weakening of the AMOC under current and future global warming[5,6]. This apparent contradiction demonstrates the need to study past archives of NADW formation and dynamics, especially those with warmer-than-modern climate boundary conditions.

The configuration of bottom water currents in the North Atlantic accounts for the highly asymmetric nature of sediment accumulation across major sub-basins, which is particularly evident south of Iceland

at ~60°N (Fig. 1b). Contourite drifts are rapidly accumulating sediment bodies formed under the influence of bottom currents, deposited in a semi-continuous process[7] (Fig. 1). The largest such deposits in the North Atlantic are the Eirik Drift, deposited by DSOW along the eastern continental margin and southern tip of Greenland, and the Björn and Gardar Drifts, deposited by branches of ISOW on the eastern flank of the Reykjanes Ridge (Fig. 1). The rapidly accumulating drifts preserve high resolution sedimentary archives of current strength, ice rafting, and climate through much of the Pleistocene (last 2.6 million years), as cored on previous Ocean Drilling Program (ODP) expeditions[8–10]. However, the most deeply buried drift sediments were not previously recovered by ocean drilling, leading to uncertainty as to the pre-Pleistocene development of this component of the NADW system.

This knowledge gap has now been filled by the recovery of sections through all of the major North Atlantic ISOW and DSOW contourite drifts to the south of Iceland. These successions were recovered in five drill sites along a latitudinal transect at ~60°N during International Ocean Discovery Programme (IODP) Expeditions 395C

and 395[11]. From west to east, these sites are U1602 (Eirik Drift; ~2710 m water depth), U1563 (a western extension of Björn Drift; ~1417 m), U1554 (centre of Björn Drift; ~1870 m), U1562 (eastern flank of Björn Drift; ~2003 m), and U1564 (centre of Gardar Drift; ~2208 m). The full thickness of the sedimentary packages at these sites was cored to basaltic basement (or, in the case of Site U1602, very close to it) (Fig. 1b). Site U1602 on Eirik Drift records the history of sedimentation from DSOW. The four eastern sites record sedimentation related to branches of ISOW, with Sites U1554, U1562 and U1564 being situated well within the modern depth range of ISOW for the Björn (≥1400 m) and Gardar drifts (~2000–2700 m), respectively[12,13]. Other sites previously drilled in the North Atlantic region provide important context for the transect (Supplementary Fig. S1; and Table S1). Specifically, ODP Site 982 (~1134 m) on the Rockall Plateau, a bathymetric high to the east of the study sites and above the influence of deep bottom currents provides a long-term non-drift control on regional pelagic sedimentation driven by biologic productivity (Fig. 1a). The Expeditions 395C and 395 drill cores provide a wealth of information relating to both the 'onset' and 'intensification' of Northern Hemisphere Glaciation which are dated to ~3.6 and ~2.7 Ma, respectively (see ref. 14 for discussion and definition of these terms).

## Results

The cored records from Expedition 395C and 395 indicate a variety of changes in sedimentation at ~3.6 Ma (Fig. 2; and Supplementary Figs. S2–S5). At Eirik Drift (Site U1602), a shift in sedimentation is seen as a reduction in carbonate content and an increase in biosilica in the rapidly accumulating drift deposit. In contrast, all investigated sites to the east of the Reykjanes Ridge transitioned from carbonate-rich sediment ranging from nannofossil ooze to silty clay with carbonate into darker, siliciclastic-dominated silty clays (Fig. 2a). Sediment accumulation rates increased from ~20–40 m/Myr before 3.6 Ma to ~100–200 m/Myr (Fig. 2b) in concert with the lithological transition. The Björn drift sites (Sites U1563, U1554, and U1562) have one or more discordances at this stratigraphic level that are evidenced in the cores by broken-up hardgrounds, glauconite pellets, grains coated in authigenic minerals, as well as reworked and oxide-stained microfossils (Supplementary Fig.S7 and Supplementary information). These reworked microfossils have been linked to resuspended lithic grain transport related to strong ISOW currents both in the Icelandic slope and the deeper parts of the Iceland Basin[12,15]. Furthermore, our age models for these sites reveal hiatuses of varying duration, which we interpret to have been caused by erosive down-cutting and a diachronous resumption of sedimentation (Fig. 2 and Supplementary Fig. S8). Such hiatuses in the northeastern Atlantic have been previously attributed to the erosional activity of Norwegian-Greenland Sea-sourced deep currents[16]. At the Gardar Drift Site U1564 the sediment record is apparently continuous, but is marked by a sharp increase in sedimentation rate and a change from sedimentary cycles that are thinner (~1 m), lighter in colour and more carbonate-rich (~20–60 wt%) to ones which are thicker (~4 m), darker and more clay-rich (~10–20 wt%) (Fig. 3; and Supplementary Figs. S7, and S9).

Cyclic changes in magnetic susceptibility of the sedimentary record at Site U1564 primarily reflect relative variations in carbonate versus detrital composition of the sediments. The biomagnetostratigraphic age control of the continuous sedimentation record (Supplementary Fig. S10) reveals that this cyclicity is paced predominantly by variations in obliquity. This cyclicity allowed us to develop a highly-resolved astronomical age model for the site (Fig. 3; and Supplementary Figs. S11–14) which dates the beginning of the major increase in sedimentation rates at 3.6 Ma, in good agreement with the available records of sedimentation rate and age of hiatuses from the Björn Drift sites which span between ~3.8–2.8 Ma. (Fig. 3, and Supplementary Figs. S13–14).

The complete stratigraphic record through the base of the Gardar Drift at Site U1564 makes it especially valuable for investigating details of the sedimentary transition, for which we use 5-cm resolution (~0.5 to 1 kyr temporal resolution) elemental ratios obtained from X-Ray Fluorescence (XRF) core scanning. Widely used proxies are (i) Ca/Fe, which reflects the relative contributions of biogenic carbonate to terrigenous sediment[17,18] (Fig. 3a); (ii) Zr/Rb, which reflects the silt component and relates to bottom current speed because Zr is enriched in the coarser grained sediment and Rb occurs mainly in clays[17,19] (Fig. 3b); and (iii) Ti/K, which reflects the southward transport of Iceland-sourced grains in this area. Sediments with high Ti concentrations are commonly linked to basaltic compositions eroded from Iceland and the Iceland-Scotland Ridge, whereas high K concentrations are mostly present in grains from more widely distributed continental (felsic) sources (Fig. 3c)[20–23]. These three ratios are shown in Fig. 3 through the time window 5–2 Ma alongside the magnetic inclination, the magnetic susceptibility record used for the construction of the astronomical age model and the obliquity tuning target.

The elemental ratios indicate a consistent pronounced transition in lithology that initiated at 3.6 Ma. While this was contemporaneous with the increase in sedimentation rate, the full change in sediment character took ~300 ky to complete. The decrease in Ca/Fe during this transition reflects the increasing dominance of terrigenous (drift) over carbonate (pelagic) sediment (Fig. 3a). The Zr/Rb proxy shows both a general increase in ratio and reduced amplitude variability suggesting an important change in the size and/or source of the eroded lithogenic grains, and therefore possibly reflecting the arrival of a new continuously strong bottom current regime (Fig. 3b). The Ti/K proxy has a similar character to Zr/Rb, mainly indicating a change in sediment source towards more basaltic grains at times of strengthened bottom currents (Fig. 3c). We note also that a second regime change in the sedimentary characteristics and cyclicity occurs at 2.7 Ma (Fig. 3), related to the well-known Northern Hemisphere glacial intensification[14]. At Eirik Drift Site U1602 the Zr/Rb shows no sustained trend or change in amplitude at 3.6 Ma, in good agreement with the lithological observations (Fig. 2a).

In summary, evidence from erosion / deposition, sedimentation rate changes and lithology all suggest that a profound change in the configuration of sediment accumulation patterns occurred across the eastern flank of the Reykjanes Ridge at 3.6 Ma. The change can be characterised as a transition from slow pelagic sedimentation to rapid contourite deposition. The contrast is more marked at the Björn Drift sites (U1563, U1554 and U1562) where widespread erosion and non-deposition defines the base of the drift. In contrast, Gardar Drift does not have a strongly erosive base at Site U1564, but pronounced increases in terrigenous-dominated sediment accumulation (exceeding 100 m/Myr) mark the onset of drift formation. This sedimentary evidence indicates a sustained increase in both southward flowing bottom current speeds and north-to-south transport, bringing more Icelandic material to the study sites at ~60 °N.

## Discussion

The combined evidence from the Expedition 395C and 395 drill cores can be explained if ISOW, currently a significant deep-water mass that dominates sedimentation patterns in the eastern North Atlantic, effectively switched on, or at least drastically intensified, at 3.6 Ma, with its influence likely growing for the following ~300 kyr. Such an interpretation assumes the entire ISOW current intensifies, and that the observed change in drift sedimentation is not related to a change in depth of ISOW[13]. The observation that the sedimentary change occurs in several sites spanning different palaeodepths over a large distance strengthens our interpretation that our observations are related to a large-scale palaeoceanographic change, and not a subtle change in the depth of ISOW. Although deep-water currents presumably flowed along similar routes for millions of years prior to the transition[24,25], they were insufficiently high or sediment laden to achieve for the accumulation rates observed after 3.6 Ma (Fig. 2b). The switch was not

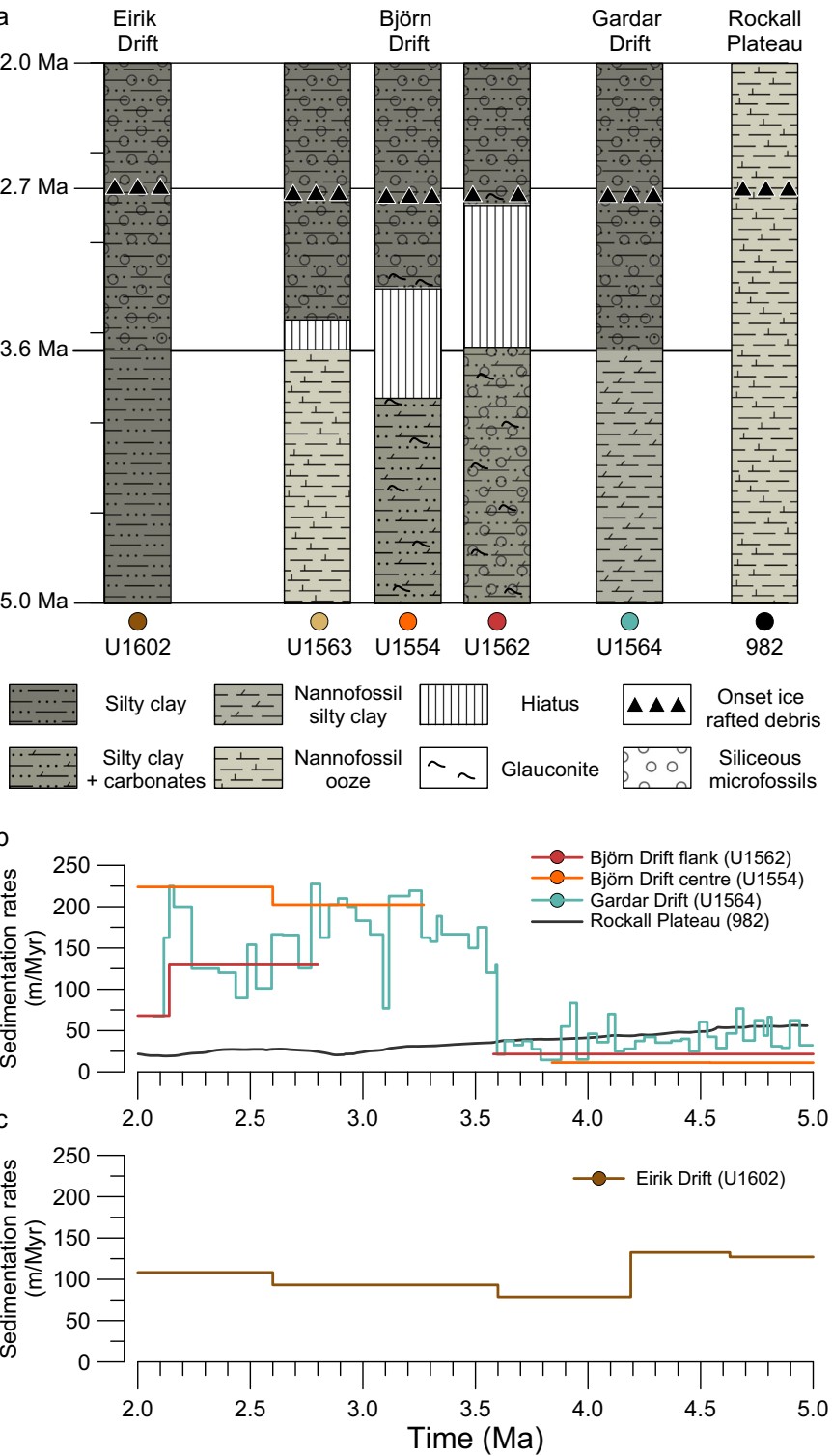

**Fig. 2 | Lithology and sedimentation rates. a** Simplified lithological columns spanning 5–2 Ma from sedimentological descriptions (Supplementary information) and physical properties (Supplementary Fig. S15). Depth-to-age conversion based on age models (from magnetostratigraphic datums) generated during Expeditions 395 and 395 C, and smoothed astronomically-tuned age model for Ocean Drilling Program (ODP) Site 982 (Supplementary information and ref. 54). Indicated are dominant lithology (colour and pattern fill), main sedimentological features (symbols), duration of hiatuses and onset of sustained presence of ice-rafted debris (IRD) (Supplementary information). **b** Sedimentation rate time series for Iceland–Scotland Overflow Water system from Expedition 395 sites and ODP Site 982 (gaps represent inferred hiatuses). Sedimentation rates for Site U1564 are from the astronomical tuning presented in this study. **c** Sedimentation rate time series at Site U1602 for the Denmark Strait Overflow Water system.

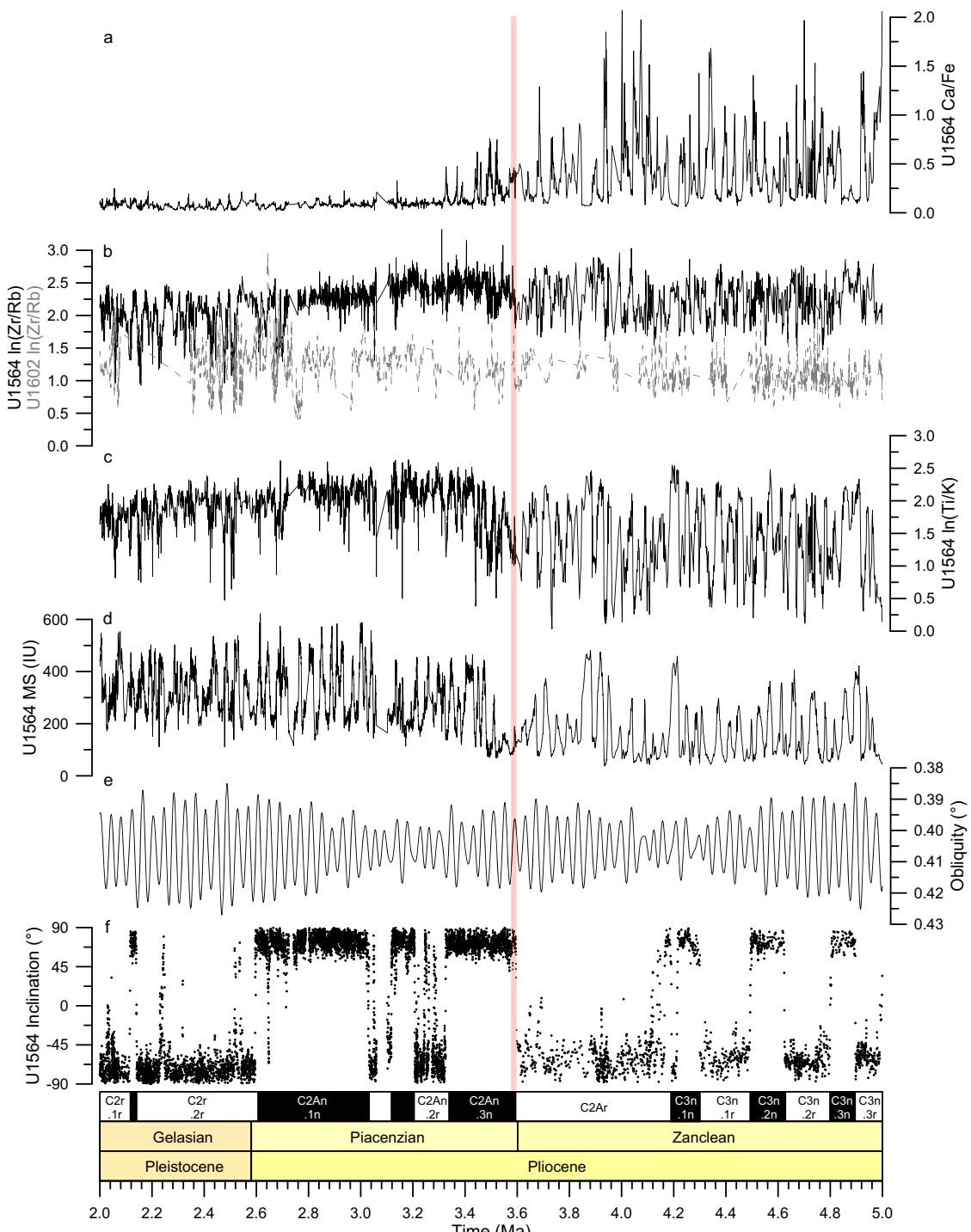

**Fig. 3 | XRF-derived elemental ratios time series and records used for U1564 age model construction. a** Ca/Fe, **b** ln(Zr/Rb) for U1564 and U1602 (dotted line), **c** ln(Ti/K), **d** Magnetic susceptibility (MS) for the spliced record. **e** Astronomical solution of Earth's obliquity[55]. **f** Palaeomagnetic inclination data (at 20 mT step) for the spliced record. All U1564 time series are plotted against ages derived from the astronomically tuned age model. Red shaded rectangle indicates ~3.6 Ma change in sedimentation.

recorded at ODP Site 982 because it is situated on a bathymetric high (~1134 m water depth) above the influence of ISOW. In contrast, our record of constantly rapid siliciclastic sedimentation at Site U1602 on Eirik Drift indicates that DSOW operated constantly throughout the Pliocene. The decrease in biogenic carbonate and increase in siliceous microfossil abundance at Site U1602 (Fig. 2a) are similar to the record previously obtained at ODP Site 646[9] located near the southern end of Eirik Drift (Fig. 1a). These lithological changes may denote a coeval

change in plankton ecology from carbonate producers (coccolitho-phorids and foraminifera) to silica producers (diatoms and radiolaria), and/or an increase in silica preservation.

An intensified ISOW at 3.6 Ma is consistent with earlier suggestions that the strongest Pliocene NADW formation occurred between ~3.6–2.7 Ma[26]. This suggestion is based on the reduced benthic $\delta^{13}C$ gradient ($\Delta\delta^{13}C$) between the North and South Atlantic Oceans. Additionally, a period of intense deep-ocean ventilation related to

strengthened Pliocene thermohaline circulation has been proposed to reach a maximum at 3.6 Ma, based on equatorial Atlantic $\delta^{13}C$ records[27]. However, that study shows a weakening in NADW immediately after 3.6 Ma lasting until ~3.4 Ma, in contrast with the period of increasing ISOW influence we propose. Another study suggests a broad-scale AMOC weakening between ~3.8–3.0 Ma based on a reduced $\delta^{18}O_{seawater}$ gradient between the North and South Atlantic Oceans[28]. Regarding the upper water column, sea surface temperature (SST) proxy records and temperature gradients from several high-latitude North Atlantic sites do show the expected signature of strengthened northwards heat transport during the Late Pliocene, while others show the opposite trend, and no sustained, unidirectional change is observed across the transition to a stronger ISOW state. High northern latitude sites such as ODP 982 (~58°N) and ODP 907 (~69°N), both record an increase in North Atlantic SST between ~3.6–3.4 Ma (ref. 29 and references therein). Similarly, a decrease in North Atlantic SST latitudinal gradient (41–58°N) is observed between 3.7–3.4 Ma[29] and could suggest stronger NAC activity. On the other hand, pronounced SST cooling between 3.65 and 3.50 Ma has been reported at ODP Site 642 on the elevated Vøring Plateau[29,30] (~67°N), and at ODP Site 610[31] (~53°N) with the latter being suggested to reflect an AMOC weakening.

The classic Panama hypothesis links the closure of the Central America Seaway (CAS) starting at ~4.6 Ma to an AMOC strengthening, and associated increase in humidity transport towards the Arctic, acting as a precondition for the growth of continental ice sheets at 2.7 Ma[27,32]. Our ISOW records shows that at least one of the main NADW components intensified within this 4.6 to 2.7 Ma time interval, indicating that crossing a climatic threshold linked to this mechanistic chain could support AMOC strengthening and the onset of Arctic glaciation at 3.6 Ma. However, issues remain with the timing of the CAS closure, which has been described to involve a broad spectrum of definitions (underwater collision to surface water exchange), as well as highly variable age estimates for its various stages ranging from ~10–2.8 Ma[27,33–37]. Additionally, modelling studies suggest that CAS closure would have led to the warming of the high Northern latitudes due to increased AMOC strength[38] which would counteract Arctic glaciation instead of promoting it. With DSOW showing no intensification and lack of Pliocene data from the Labrador Sea, extrapolating ISOW results to larger scale changes of the AMOC activity should be done with caution and integrated with basin-wide changes in surface circulation and deep-water ventilation.

Other potential forcing mechanisms for the ISOW initiation include gradual changes in ocean basin topography and gateways caused by plate boundary reorganisation. Among relevant gateway changes, it has been suggested that the Iceland mantle plume transitioned to a lower buoyancy flux mode between ~6–2.5 Ma, reducing regional dynamic (i.e., sub-plate) support and lowering sill depth across both the Denmark Strait and Iceland–Scotland Ridge[39–41]. This proposed reduction in mantle plume buoyancy flux began too early to explain the abrupt ISOW intensification at ~3.6 Ma, although it could perhaps have been a precondition for ISOW formation. Similarly, both the Bering[42] and Fram[43] straits were already open long before 3.6 Ma, with the latter allowing deep outflow from the Arctic to the Nordic Seas.

A possible climatic driver of the ISOW intensification is Arctic cooling itself, which could have been independent of the CAS closure and perhaps driven by a broad-scale greenhouse gas decline[44]. This $CO_2$ decline may have cooled the Norwegian Sea towards a tipping point that accelerated down-welling and ISOW, which then reinforced cooling. Conclusive testing of such hypotheses requires modelling of Atlantic palaeobathymetry combined with more detailed and widespread proxy records, especially SST and salinity records from the North Atlantic, as well as reconstruction of the Labrador Sea Water as a remaining major contributing component of NADW. Future multi-proxy work across the newly recovered Expedition 395C and 395

transect will add geographic and temporal coverage to help resolve these issues.

In summary, the first sedimentary records to penetrate through all the major North Atlantic contourite drifts firmly establish the initiation of a strong ISOW at ~3.6 Ma against the sustained long-term operation of DSOW. In addition, we note a close coincidence in timing of ISOW intensification with the onset of Northern Hemisphere Glaciation. More generally, we demonstrate that the western (DSOW) and eastern (ISOW) components of NADW have contrasting histories, underlining the need to regard them as potentially independent actors in driving and responding to climate change.

## Methods
### Sedimentology
Lithologic descriptions were based on visual core inspection and smear slide analysis. The principal lithologic name was assigned on the basis of the relative abundance of terrigenous clastic and biogenic grains. Sediments with > 50 percent terrigenous grains were classified based on the dominant terrigenous grain size, indicated by the name. A preceding modifying name was used if a different grain size component contributes at least thirty percent of the terrigenous sediment. A component with 10 to 29 percent is indicated by addition of the modifier 'with …', e.g., silty clay with carbonate. Minor sedimentary components ( < 10 percent) were not included in the name. Sediments with > 50 percent biogenic material were classified as oozes, modified by the most abundant specific biogenic component or carbonate/biosilica if they were both present. Additional components comprising 30 to 49 percent of the sediment preceded the primary ooze name. As with the terrigenous sediment,'with' was used for a minor (10 to 29 percent) component. For example, silty clay with carbonate contains 70 percent silty clay and 10–29 percent carbonate tests, which could be as low as eight percent nannofossils and two percent foraminifera.

### Physical properties
Magnetic susceptibility (MS) was measured point-wise on split core sections using a Bartington Instruments MS2E point sensor (2.0 or 2.5 cm resolution). MS measurements are reported as Instrumental Units (IU) as the core mass is not measured prior to data acquisition and the internal volume cannot be determined until core is split. The correction for the volume does not change the order of magnitude of the measured susceptibility values, so the results are comparable with the susceptibility measured by the palaeomagnetism equipment which reports data using SI units.

### XRF scanning
X-Ray fluorescence (XRF) derived elemental ratios were calculated from elemental intensities, which were measured in core-section half using the Avaatech XRF Core Scanners of the IODP Gulf Coast Repository for Site U1564 and the Institute of Geophysics and Planetary Physics at Scripps Institution of Oceanography for Site U1602. For each core-section, three energies (10, 30, and 50 keV) were measured for the spliced interval of Holes U1564C-D-E and U1602D at a resolution of 5 cm. All samples that had positive Ar counts as measured in the 10 keV run were removed as Ar counts larger than zero were interpreted as an indication for bad contact between the core surface and XRF sensor. The Ar correction was not applied for the Site U1602 record as the detector used has a different sensitivity compared to the instruments used for scanning the other sites. The natural logarithm (ln) of three elemental ratios, namely Ca/Ti, Zr/Rb and Ti/K, are used as indicators of the relative contribution of carbonate to terrigenous material, bottom current strength and mineral grain sources, respectively. The ratio of Ti/Ca has been widely used to infer the relative dominance of terrestrial processes over carbonate production in the water column[17,18]. In cases where sediment is current sorted, as in drift and glaciomarine deposits, Zr/Rb and Zr/Al have been shown to reflect bottom current

strength and grain size (e.g., refs. [17],[19]). In our study area, the elemental ratios between Ti and K have been shown to reflect the grain source of basaltic, Iceland-derived sediment grains, and to infer stronger ISOW during interstadials (e.g., ref. [23] and references therein).

### Downhole logging

Downhole magnetic susceptibility (MSS-B) and natural gamma radiation (HNGS tool) were collected with the triple combination tool string in open hole after coring operations had concluded. The MSS-B tool incorporates a single-coil electric inductor sensor that when positioned against the borehole wall is capable of measuring inductance as affected by the volume susceptibility of the formation over the volume of investigation. The MSS-B tool provided ~40 cm vertical resolution measurements with a ~20 cm depth of horizontal investigation. The magnetic susceptibility is recorded in Instrument Units (IU) every 2.5 cm. The MSS-B tool is designed to have a similar response function as the Barington Instruments whole round sensor used on the whole round multisensor core logger to allow meaningful comparison between the two measurements as well as with the susceptibility measured by the palaeomagnetism equipment. The HNGS tool uses two bismuth-germanate (BGO) scintillation detectors to measure the natural gamma radiation of a formation (HSGR), measured in American Petroleum Institute units (gAPI). The HNGS tool provides data at a vertical resolution of 30.48 cm and has a depth of investigation of 61 cm. The data were collected at a 15 cm sampling rate.

### Palaeomagnetism

Shipboard palaeomagnetic investigations were used to determine directions of natural remanent magnetisation components and to identify downhole variation of magnetic polarities. Palaeomagnetic measurements were carried out on archive half-core sections at a resolution of 2.5 cm with in-line stepwise alternating field (AF) demagnetisation generally up to 25 or 30 mT on a 2 G Enterprises Model-760R-4K superconducting rock magnetometer equipped with direct-current SQUIDs and an in-line, automated AF demagnetiser. Additionally, oriented discrete samples (7 cm³ Natsuhara-Giken cubes; 8 cm³ cut cubes) were collected from working-half sections to confirm the polarity changes. Stepwise AF demagnetisation was imparted using a DTECH (model D-2000) AF demagnetiser up to 45 mT for the low-coercivity samples, and up to 100 mT for high-coercivity samples. The remaining magnetisation was measured on a spinner magnetometer (AGICO, model JR-6A) after each step. In most cases, after the removal of a drilling overprint up to 10 mT, a stable and primary magnetisation was successfully isolated. For determining the magnetic polarities in each hole, inclination data at the maximum demagnetisation step were used from archive-half sections, while discrete sample data were analysed by principal component analysis using a Fisherian statistic[45] to isolate the characteristic remanent magnetisation[46] using the Puffin-Plot software[47]. The magnetostratigraphy was inferred by correlating the magnetic polarities with the reference geomagnetic polarity time scale using Table 5.3 (p. 166) of ref. [48].

### Biostratigraphy

The biostratigraphy was based on calcareous nannofossils and planktonic foraminifer assemblages taken from core catcher and in-section samples. Analyses focused on the identification of biostratigraphic horizons (biohorizons) that have been previously assigned absolute ages based on calibrations from other sites, referenced to the palaeomagnetic reversal sequence or astronomical timescales. Tables of biohorizon depths and calibrated biohorizon ages are given in the Supplementary Information. Calcareous nannofossil biohorizons were determined using standard smear slide preparations, which were examined with transmitted light microscopy. Biohorizon age assignments follow the latest compilation of ref. [49]. Planktonic foraminifers were extracted from sample volumes of 10 or 20 cm³, using standard disaggregation techniques and then washing over 63 μm sieves. Dried residues of the > 63 μm were examined using a binocular stereomicroscope. Planktonic foraminifera biohorizons, based principally on previous high latitude biozonations of refs. [50],[51] with elements of the standard (sub)tropical biozonation of ref. [52] as updated by ref. [49]). Where necessary, planktonic foraminifera biohorizon ages were recalibrated to the GPTS using Table 5.3 (p. 166) of ref. [48] and ref. [53] taking into account the errors produced by the sampling resolution of the biostratigraphy and palaeomagnetics of the original data.

### Age models

For each site, the palaeomagnetic and biostratigraphic data were collated on the common geological time scale using Table 5.3 (p. 166) of ref. [48] modified to include palaeomagnetic reversal ages of ref. [53]. Age models (Supplementary information) for the sedimentary successions of at least one hole at each site were constructed using shipboard palaeomagnetic and biostratigraphic age constraints, which were interpreted alongside information from sedimentology, physical properties and seismic lines. Age–depth plots were used to determine a series of age model 'tie points' between which linear interpolations were made, as well as identifying the depth levels of possible hiatuses or zones of mixed sedimentation and/or slumping. Where the palaeomagnetic records are relatively clear and unambiguous, and consistent with the biostratigraphic data, each well-defined palaeomagnetic reversal was used as a tie point.

### Cyclostratigraphy

Construction of an astronomical age model was based on the correlation of the relative variability of the magnetic susceptibility signal with the LR04 stack (ref. [54]) and astronomical solutions of precession and obliquity from ref. [55]. The relative phase relationships were determined based on the stratigraphic correlation of the MS profile of the Site U1564 splice with the overlapping stratigraphy of the well-studied lowest hole of ODP Hole 983 C (ref. [56]), further supported by palaeomagnetic data. The phase relationships are as follows: high values in MS corresponding with more positive benthic foraminifera oxygen isotope values (glacial periods) and relative minima in obliquity. See Supplementary information for a detailed description of the tuning process, tuning figures (Supplementary Figs. S11–S14) and Site U1564 astronomically tuned age-depth table (Supplementary information).

## Data availability

The shipboard core and wireline logging data have been deposited in the International Ocean Discovery Programme, JOIDES Resolutions Science Operator LIMS database under public accession code https://web.iodp.tamu.edu/LORE/. The XRF elemental data for Site U1602 have been deposited in the public Zenodo database under accession code https://doi.org/10.5281/zenodo.15065343. The IRD and glauconite grain estimates based on palaeontological residues, and U1564 astronomical tuning tie points generated in this study are provided as Supplementary data.

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

## Acknowledgements

This research used samples and data provided by the International Ocean Discovery Programme (IODP). We thank the *JOIDES Resolution* Science Operator technical staff and crew of the *JOIDES Resolution* who sailed on Expeditions 384, 395C, and 395 for their tremendous efforts throughout and following the COVID-19 pandemic. Without their resolve and expertise, this research would not have been possible. This work was supported by European Research Council (ERC) Advanced Grant AstroGeo-885250 (J. Laskar; M.S.); IODP-France (A.B., G.P., M.S.); the micropaleontology lab of Geocentrum (J. Henderiks, UU); the Katharina von Salis fellowship and Anna Maria Lundin's travel grant (B.Th.K.); the EUTOPIA Science and Innovation Fellowship Programme and European Union Horizon 2020 programme under the Marie Skłodowska Curie grant agreement No945380 (B.Th.K.); UK Natural Environment Research Council (NERC) grants NE/Y001745/1 and NE/W007002/1 (P.N.P., B.W.); National Science Foundation (NSF) awards OCE-1326927 and OCE2412279 (L.J.L.); NERC NE/Y003772/1 (T.D.J.); Australian Research Council Special Research Initiative, Australian Centre for Excellence in Antarctic Science Project Number SR200100008 (K.H.); NERC NE/W004887/1 and NE/Y002393/1 (D.D.M.); NERC NE/W002167/1 (B.J.M.); MUR for ECORD-IODP Italia (A.D.C., S.S.); NERC NE/W007150/1 (N.J.W.); NSF award OCE-2238290 (R.P.-T.); United States Science Support Programme (S.R.H., Y.C., J.P.D., D.D., D.E.E., S.A.F., H.E.I., C.J., D.E.L., M.R.L., S.OC., Y.R., and R.P.-T.).

## Author contributions

R.P.-T., A.B., N.W., B.J.M., and D.E.E. contributed as proponents to the IODP proposal. R.P.-T., A.B., and L.J.L. led IODP Expeditions 395 and 395C. M.S. and B.Th.K. conceived the study, performed the data compilations, interpreted the data, and wrote the manuscript in consultation with P.N.P., A.B., S.R.H., L.J.L., T.D.J., and R.P.-T. M.S., B.Th.K., S.R.H., P.N.P., T.D.J., T.S., A.D.C., S.A.F., and S.S. conducted the biostratigraphic and magnetostratigraphic analysis, generated the stratigraphic correlation, and constructed the age models. H.E.I., C.E.J., S.L., D.E.L., S.E.M., S.O., and T.T. provided the sedimentological analysis and lithological descriptions. E.H. and M.M. helped with post-processing the XRF scanning records. M.S., D.D., K.H., D.D.M., and N.J.W. measured the physical properties of the sediments. Y.C., J.P.D. D.E.E., S.L., M.R.L., B.J.M., G.P., S.-P.Q., Y.R., B.S.W., N.J.W., T.W., and A.Y.Y. contributed to data acquisition, discussions, and manuscript editing.

## Competing interests

The authors declare no competing interests.

## Additional information

¹IMCCE, Observatoire de Paris, 77 avenue Denfert-Rochereau, Paris, France. ²Geology, School of Natural Sciences, Trinity College Dublin, The University of Dublin, College Green, Dublin, Ireland. ³Department of Earth Sciences, Uppsala University, Villavägen 16, 752 36, Uppsala, Sweden. ⁴Archaeology, Environmental changes & Geo-Chemistry, Vrije Universiteit Brussel, Pleinlaan 2, Brussels, Belgium. ⁵Department of Earth Sciences, University College London, Gower Street, London, United Kingdom. ⁶Geo-Ocean, Centre National de la Recherche Scientifique (CNRS), Institut Universitaire Européen de la Mer, Rue Dumont d'Urville, Plouzané, France. ⁷Department of Earth and Environmental Sciences, Columbia University, New York, NY, USA. ⁸Lamont-Doherty Earth

Observatory, Columbia University, Palisades, NY, USA. [9]International Ocean Discovery Program, Texas A&M University, 1000 Discovery Drive, College Station, TX, USA. [10]School of Geography, Earth and Environmental Sciences, University of Birmingham, Birmingham, United Kingdom. [11]Department of Earth and Environmental Studies, Montclair State University, 1 Normal Ave., Montclair, NJ, USA. [12]Istituto Nazionale di Geofisica e Vulcanologia, Via di Vigna Murata 605, Roma, Italy. [13]Earth, Atmosphere and Environment, Northern Illinois University, 180 Stadium Drive, DeKalb, IL, USA. [14]College of Earth Ocean and Atmospheric Sciences, Oregon State University, 1500 SW, Jefferson Way, Corvallis, OR, USA. [15]Department of Earth Sciences, School of Ocean & Earth Science & Technology, University of Hawai'i at Mānoa, 1680 East–West Road, Honolulu, HI, USA. [16]School of Earth, Environment & Sustainability, Georgia Southern University, Statesboro, GA, USA. [17]Institute of Marine and Antarctic Studies (IMAS), College of Science and Engineering, University of Tasmania, Hobart, Australia. [18]School of Geography, Geology and the Environment, University of Leicester, Leicester, United Kingdom. [19]Department of Earth Sciences Binghamton University, Binghamton, NY, USA. [20]Atmosphere and Ocean Research Institute, The University of Tokyo, Kashiwa, Chiba, Japan. [21]Department of Earth and Environmental Sciences, Boston College, 140 Commonwealth Avenue, Chestnut Hill, MA, USA. [22]Bigelow Laboratory for Ocean Sciences, 60 Bigelow Drive, East Boothbay, ME, USA. [23]Department of Earth, Ocean and Ecological Sciences, University of Liverpool, 4 Brownlow Street, Liverpool, United Kingdom. [24]Department of Geography and Environmental Sciences, Northumbria University, Newcastle upon Tyne, United Kingdom. [25]Institute of Geophysics & Planetary Physics, Scripps Institution of Oceanography, University of California, San Diego, La Jolla, CA, USA. [26]Ocean Bio-Geoscience Group, National Oceanography Centre, Southampton, United Kingdom. [27]Department of Earth and Environmental Sciences, Wesleyan University, 265 Church Street, Middletown, CT, USA. [28]Complex Fluids and Reservoirs Laboratory, University of Pau and Pays de l'Adour, Avenue De l'Université, Pau, France. [29]Southern Marine Science and Engineering Guangdong Laboratory (Guangzhou), Guangzhou, China. [30]Department of Marine and Coastal Sciences, Rutgers, The State University of New Jersey, 71 Dudley Road, New Brunswick, NJ, USA. [31]Department of Engineering and Geology, University of Chieti-Pescara, Via dei Vestini 31, Chieti, Italy. [32]SUGAR, X-star, Japan Agency for Marine-Earth Science and Technology (JAMSTEC), Yokosuka, Japan. [33]Geosciences Division, National Centre for Polar and Ocean Research (NCPOR), Vasco-da-Gama, Goa, India. [34]Bullard Laboratories, Department of Earth Sciences, University of Cambridge, Madingley Road, Cambridge, United Kingdom. [35]Ocean College, Zhejiang University, Zhoushan, China. [36]Guangzhou Institute of Geochemistry, Chinese Academy of Sciences, 511 Kehua Street Tianhe District, Guangzhou, Guangdong, China. [37]These authors contributed equally: Matthias Sinnesael, Boris-Theofanis Karatsolis. ✉e-mail: sinnesam@tcd.ie

