## [Transparent Peer Review file · Nature Communications]

Onset of strong Iceland-Scotland Overflow Water 3.6 million years ago

Corresponding Author: Dr Matthias Sinnesael

Version 0:

Reviewer comments:

Reviewer #1

(Remarks to the Author)
Dear Editor and Author(s),

Thank you for the opportunity to review the manuscript entitled "Onset of strong Iceland-Scotland Overflow Water 3.6 million years ago" by Dr. Sinnesael and co-authors. This manuscript has now undergone two rounds of review. The authors have clearly taken previous feedback into account and made significant improvements in key areas.

The manuscript's title has changed from previous versions, and I now believe it best reflects the content. The latest version of the manuscript clarifies the study's implications and reduces speculative interpretation by narrowing the focus to ISOW and its use in constraining NADW. The use of core data for multi-proxy analysis is also commendable, and previous concerns raised by past reviewers have been addressed with clear rebuttals.

Overall, I recommend acceptance of the manuscript in its current form, and I look forward to seeing it published.

Best wishes,
Dr. Harya Nugraha

Reviewer #2

(Remarks to the Author)
The authors satisfactorily addressed my comments, and I think the manuscript is almost ready for publication. I just have a minor point that the authors need to revise.

In lines 195-199 the authors aim to support their finding of an intensified ISOW at around 3.6 Ma with previous findings of NADW strengthening. However, in lines 178-179 it is specified that ISOW strengthening occurred during 3.6-3.3 Ma and this time period does not match with the study mentioned in lines 198-199 (Haug and Tiedemann, 1998). In this study, an increased NADW formation is proposed from about 4.6 Ma onwards, reaching a maximum at about 3.6 Ma, and followed by a decrease until about 3.4 Ma. Hence, this discrepancy needs to be better explained.

REVIEWERS' COMMENTS

Reviewer #1 (Remarks to the Author):

Dear Editor and Author(s),

Thank you for the opportunity to review the manuscript entitled "Onset of strong Iceland-Scotland Overflow Water 3.6 million years ago" by Dr. Sinnesael and co-authors. This manuscript has now undergone two rounds of review. The authors have clearly taken previous feedback into account and made significant improvements in key areas.

The manuscript's title has changed from previous versions, and I now believe it best reflects the content. The latest version of the manuscript clarifies the study's implications and reduces speculative interpretation by narrowing the focus to ISOW and its use in constraining NADW. The use of core data for multi-proxy analysis is also commendable, and previous concerns raised by past reviewers have been addressed with clear rebuttals.

Overall, I recommend acceptance of the manuscript in its current form, and I look forward to seeing it published.

Best wishes,
Dr. Harya Nugraha

We thank Dr. Nugraha for reviewing the manuscript, and the general appreciation.

Reviewer #2 (Remarks to the Author):

The authors satisfactorily addressed my comments, and I think the manuscript is almost ready for publication. I just have a minor point that the authors need to revise.

In lines 195-199 the authors aim to support their finding of an intensified ISOW at around 3.6 Ma with previous findings of NADW strengthening. However, in lines 178-179 it is specified that ISOW strengthening occurred during 3.6-3.3 Ma and this time period does not match with the study mentioned in lines 198-199 (Haug and Tiedemann, 1998). In this study, an increased NADW formation is proposed from about 4.6 Ma onwards, reaching a maximum at about 3.6 Ma, and followed by a decrease until about 3.4 Ma. Hence, this discrepancy needs to be better explained.

We thank Reviewer #2 for reviewing another version of this the manuscript. We rephrase this part of the discussion accordingly:

“Additionally, a period of intense deep-ocean ventilation related to strengthened Pliocene thermohaline circulation has been proposed to reach a maximum at 3.6 Ma, based on equatorial Atlantic $\delta^{13}\text{C}$ records²⁷. However, this study shows a weakening in NADW immediately after 3.6 Ma lasting until ~ 3.4 Ma, in contrast with the period of increasing ISOW influence we propose. Another study suggests a broad-scale AMOC weakening between ~ 3.8 -3.0 Ma based on a reduced $\delta^{18}\text{O}_{\text{seawater}}$ gradient between the North and South Atlantic Oceans²⁸. Regarding the upper water column,...”